# Effective Learning with Node Perturbation in Multi-Layer Neural Networks

## Abstract

Backpropagation (BP) remains the dominant and most successful method for training parameters of deep neural network models. However, BP relies on two computationally distinct phases, does not provide a satisfactory explanation of biological learning, and can be challenging to apply for training of networks with discontinuities or noisy node dynamics. By comparison, node perturbation (NP) proposes learning by the injection of noise into network activations, and subsequent measurement of the induced loss change. NP relies on two forward (inference) passes, does not make use of network derivatives, and has been proposed as a model for learning in biological systems. However, standard NP is highly data inefficient and unstable due to its unguided noise-based search process. In this work, we investigate different formulations of NP and relate it to the concept of directional derivatives as well as combining it with a decorrelating mechanism for layer-wise inputs. We find that a closer alignment with directional derivatives together with input decorrelation at every layer strongly enhances performance of NP learning with large improvements in parameter convergence and much higher performance on the test data, approaching that of BP. Furthermore, our novel formulation allows for application to noisy systems in which the noise process itself is inaccessible.

## 1 Introduction

Backpropagation (BP) is the workhorse of modern artificial intelligence. It provides an efficient way of performing multi-layer credit assignment, given a differentiable neural network architecture and loss function (Linnainmaa, 1970). Despite BP's successes, it requires an auto-differentiation framework for the backward assignment of credit, introducing a distinction between a forward, or inference phase, and a backward, or learning phase, increasing algorithmic complexity and impeding implementation in (neuromorphic) hardware (Kaspar et al., 2021; Zenke & Neftci, 2021). Furthermore, BP has long been criticized for its lack of biological detail and plausibility (Grossberg, 1987; Crick, 1989; Lillicrap et al., 2016), with significant concerns again being the two separate forward and backward phases, but also its reliance on gradient propagation and the non-locality of the information required for credit assignment.

Alternative algorithms have been put forth over the years, though their inability to scale and difficulty in achieving levels of performance comparable to BP have held back their use. Many focus either on greedy local learning rules or on random feedback matrices and projections, which are deemed more plausible than symmetric feedback weights (Dellaferrera & Kreiman, 2022; Frenkel et al., 2021; Kohan et al., 2018; Hinton, 2022)

The category of algorithms we focus on in this work are perturbation algorithms, such as node perturbation (NP) (Dembo & Kailath, 1990; Cauwenberghs, 1992), which require no feedback connectivity at all. In NP, node activations are perturbed by a small amount of random noise. Weights are then updated to produce the perturbed activations in proportion to the degree by which the perturbation improved performance. This method requires a measure of the loss function being optimized on a network both before and after the inclusion of noise. Such an approach is appealing because it leverages the same forward pass twice, rather than relying on two computationally distinct phases. It also does not require non-local information other than the global performance signal.

Despite these benefits, Hiratani et al. (2022) demonstrate that NP is extremely inefficient compared to BP, requiring two to three orders of magnitude more training cycles, depending on network depth

and width. In addition, they found training with NP to be unstable, in many cases due to exploding weights. Another phenomenon uncovered in their work is that the covariance of NP's updates is significantly higher than that of BP, which can mathematically be described as an effect mediated by correlations in the input data.

Another proposed approach, referred to as weight perturbation (WP), is to inject noise directly into the weights of a network, rather than the activations (Werfel et al., 2003; Fiete & Seung, 2006). This method shows similarities to evolutionary algorithms and can outperform NP in special cases (Züge et al., 2021). A downside to WP is that there are many more weights than nodes in neural networks, making the exploration space larger thus slowing down learning. In fact, both NP and WP lag far behind BP in terms of efficiency. This is to be expected, however, as noise perturbations conduct a random search through parameter space, rather than being gradient directed.

In this work, we put forward three contributions. First, we reframe the process of node perturbation together with the subsequent measurement of the output loss change in terms of directional derivatives. This provides a more solid theoretical foundation for node perturbation and, consequently, a different update rule, which we refer to as iterative node perturbation. Directional derivatives have been related to NP before in the context of forward gradient learning (Baydin et al., 2022; Ren et al., 2022). This approach is different from ours in that the perturbations are used to estimate gradients, but node-perturbed forward passes are not performed network-wide, making these approaches unsuitable for use in true noisy systems.

Second, we introduce an approximation to iterative node perturbation, referred to as activity-based node perturbation, which is more efficient and has the additional advantage that it can be implemented in noisy systems such as imprecise hardware implementations (Gokmen, 2021) or biological systems (Faisal et al., 2008), where the noise itself is not directly measurable.

Third, we propose to use a decorrelation method, first described by Ahmad et al. (2023), to debias layer-wise activations and thereby achieve faster learning. Because NP-style methods directly correlate perturbations of unit activities with changes in a reward signal, decorrelation of these unit activities helps to eliminate confounding effects, making credit assignment more straightforward. In addition, as demonstrated by Hiratani et al. (2022), correlations in the input data lead to more bias in NP's updates by increasing their covariance. By combining decorrelation with the different NP methods, we find that it is possible to achieve orders of magnitude increase in model convergence speed, with performance levels rivaling networks trained by BP in certain contexts.

## 2 METHODS

### 2.1 NODE PERTURBATION AND ITS FORMULATIONS

Let us define the forward pass of a fully-connected neural network, with $L$ layers, such that the output of a given layer, $l \in 1, 2, \ldots, L$ is given by $\mathbf{x}_l = f(\mathbf{a}_l)$, where $\mathbf{a}_l = \mathbf{W}_l \mathbf{x}_{l-1}$ is the pre-activation with weight matrix $\mathbf{W}_l$, $f$ is the activation function and $\mathbf{x}_l$ is the output from layer $l$. The input to our network is therefore denoted $\mathbf{x}_0$, and the output $\mathbf{x}_L$. We consider learning rules which update the weights of such a network of the form

$$\mathbf{W}_l \leftarrow \mathbf{W}_l - \eta \Delta \mathbf{W}_l$$

where $\eta$ is a small constant learning rate and $\Delta \mathbf{W}_l$ is a parameter update direction for a particular algorithm. Recall that the regular BP update is given by

$$\Delta \mathbf{W}_l = \mathbf{g}_l \mathbf{x}_{l-1}^\top \tag{1}$$

with $\mathbf{g}_l = \frac{\partial \mathcal{L}}{\partial \mathbf{a}_l}$ the gradient of the loss $\mathcal{L}$ with respect to the layer activations $\mathbf{a}_l$. Our aim is (1) to derive gradient approximations which have better properties than standard node perturbation and (2) to show that decorrelation massively improves convergence performance for any of the employed learning algorithms. In the following, we consider weight updates relative to a single input sample $\mathbf{x}_0$. In practice, these updates are averaged over mini-batches.

### 2.1.1 TRADITIONAL NODE PERTURBATION

In the most common formulation of NP, noise is injected into each layer's pre-activations and weights are updated in the direction of the noise if the loss improves and in the opposite direction if it worsens.

Two forward passes are required: one clean and one noise-perturbed. During the noisy pass, noise is injected into the pre-activation of each layer to yield a perturbed output

$$\tilde{\mathbf{x}}_l = f\left(\tilde{\mathbf{a}}_l + \boldsymbol{\epsilon}_l\right) = f\left(\mathbf{W}_l \tilde{\mathbf{x}}_{l-1} + \boldsymbol{\epsilon}_l\right) \tag{2}$$

where the added noise $\boldsymbol{\epsilon}_l \sim \mathcal{N}(\mathbf{0}, \sigma^2 \mathbf{I}_l)$ is a spherical Gaussian perturbation with no cross-correlation and $\mathbf{I}_l$ is an $N_l \times N_l$ identity matrix with $N_l$ the number of nodes in layer $l$. Note that this perturbation has a cumulative effect on the network output as each layer's perturbed output $\tilde{\mathbf{x}}_l$ is propagated forward through the network, resulting in layers deeper in the network being perturbed by more than just their own added noise.

Having defined a clean and noise-perturbed network-pass, we can measure a loss differential for a NP-based update. Supposing that the loss $\mathcal{L}$ is measured using the network outputs, the loss difference between the clean and noisy network is given by $\delta\mathcal{L} = \mathcal{L}(\tilde{\mathbf{x}}_L) - \mathcal{L}(\mathbf{x}_L)$, where $\delta\mathcal{L}$ is a scalar measure of the difference in loss induced by the addition of noise to the network. Given this loss difference and the network's perturbed and unperturbed outputs, we compute a layer-wise learning signal by replacing the gradient $\mathbf{g}_l$ in Eq. 1 with a term consisting of the loss difference and a normalized noise vector:

$$\Delta\mathbf{W}_l^{\text{NP}} = \delta\mathcal{L} \frac{\boldsymbol{\epsilon}_l}{\sigma^2} \mathbf{x}_{l-1}^\top. \tag{3}$$

### 2.1.2 ITERATIVE NODE PERTURBATION

Though mathematically simple, the traditional NP approach described above has a number of drawbacks. One of the biggest drawbacks is that each layer's noise impact upon the loss is confounded by additional noise in previous and following layers which is unaccounted for. Appendix A describes how correlations arise between layers. Furthermore, a precise relationship between the traditional NP formulation and gradient descent is missing.

In the following, we develop a more principled approach to node perturbation. Our goal is to determine the gradient of the loss with respect to the pre-activations in a layer $l$ by use of perturbations. To this end, we consider the partial derivative of the loss with respect to the pre-activation $a_l^i$ of unit $i$ in layer $l$ for all $i$. We define a perturbed state as $\tilde{\mathbf{x}}_k(h) = f\left(\tilde{\mathbf{a}}_k + h\mathbf{m}_k\right)$ with $h$ an arbitrary scalar and binary vectors $\mathbf{m}_k = \mathbf{e}_i$ if $k = l$ with $\mathbf{e}_i$ a standard unit vector and $\mathbf{m}_k = \mathbf{0}$ otherwise. We may now define the partial derivatives as

$$(\mathbf{g}_l)_i = \lim_{h \to 0} \frac{\mathcal{L}(\tilde{\mathbf{x}}_L(h)) - \mathcal{L}(\mathbf{x}_L)}{h}.$$

This suggests that node perturbation can be rigorously implemented by measuring derivatives using perturbations $h\mathbf{m}_k$ for all units $i$ individually in each layer $l$. However, this would require as many forward-passes as there exist nodes in the network, which would be extremely inefficient.

An alternative approach is to define perturbations in terms of directional derivatives. Directional derivatives measure the derivative of a function based upon an arbitrary vector direction in its dependent variables. However, this can only be accomplished for a set of dependent variables which are individually independent of one-another. Thus, we cannot compute such a measure across an entire network. We can, however, measure the directional derivative with respect to a specific layer via a perturbation given by

$$\tilde{\mathbf{x}}_k(h) = f\left(\tilde{\mathbf{a}}_k + h\mathbf{v}_k\right)$$

where $\mathbf{v}_k \sim \mathcal{N}(\mathbf{0}, \sigma^2 \mathbf{I}_k)$ if $k = l$ and $\mathbf{v}_k = \mathbf{0}$ otherwise. Here, $\sigma^2$ is a scalar, equivalent to the noise variance. Given this definition, we can precisely measure a directional derivative with respect to the activities of layer $l$ in our deep neural network, in vector direction $\mathbf{v} = (\mathbf{v}_1, \ldots, \mathbf{v}_L)$ as

$$\nabla_{\mathbf{v}}\mathcal{L} = \lim_{h \to 0} \frac{\mathcal{L}\left(\tilde{\mathbf{x}}_L(h)\right) - \mathcal{L}\left(\mathbf{x}_L\right)}{h||\mathbf{v}||}$$

where the directional derivative is measured by a difference in the loss induced in the vector direction $\mathbf{v}$, normalized by the vector length $||\mathbf{v}||$ and in the limit of infinitesimally small perturbation. The normalization ensures that the directional derivative is taken with respect to unit vectors. Note that this derivative is only being measured for layer $l$ as for all other layers the perturbation vector is composed of zeros.

As derived in Appendix B, by averaging the directional derivative across samples of the vector $\mathbf{v}$, in the limit $h \to 0$, we exactly recover the gradient of the loss with respect to a specific layer, such that

$$\mathbf{g}_l = N_l \left\langle \nabla_{\mathbf{v}} \mathcal{L} \frac{\mathbf{v}_l}{||\mathbf{v}||} \right\rangle_{\mathbf{v}} . \tag{4}$$

Equation 4 allows us to measure the gradient of a particular layer of a deep neural network by perturbation. Instead of averaging over multiple noise vectors, we use a sample-wise weight update and average over mini-batches instead. Fixing $h$ at a small value and incorporating it into the scale of the noise vector $\mathbf{v}$, we obtain weight update

$$\Delta \mathbf{W}_l^{\mathrm{INP}} = N_l \, \delta \mathcal{L} \frac{\mathbf{v}_l}{||\mathbf{v}||^2} \, \mathbf{x}_{l-1}^\top \tag{5}$$

where we used $\nabla_{\mathbf{v}} \mathcal{L} = \delta \mathcal{L} / ||\mathbf{v}||$. We refer to this method as iterative node perturbation (INP).

### 2.1.3 ACTIVITY-BASED NODE PERTURBATION

Though theoretically grounded, INP incurs significant overhead compared to the direct noise-based update method. Specifically, in order to precisely compute directional derivatives, noise must be applied in a purely layer-wise fashion, requiring a high degree of control over the noise injection. This is less biologically plausible and functionally more challenging implementation with less potential for scaling. Furthermore, the number of forward passes required to compute such an update for a whole network for INP is then the number of layers plus one, $L + 1$, potentially many times larger than the two passes required for NP.

To balance learning speed and computational cost we instead approximate the directional derivative across the whole network simultaneously. This involves assuming that all node activations in our network are independent and treating the entire network as if it were a single layer. This can be achieved by, instead of measuring and tracking the injected noise alone, measuring the state difference between the clean and noisy forward passes of the whole network. Note that this is a significant departure from traditional node perturbation in which access to the underlying noise source is required. Concretely, taking the definition of the forward pass given by NP in Eq. 2, we define

$$\Delta \mathbf{W}_l^{\mathrm{ANP}} = N \, \delta \mathcal{L} \frac{\delta \mathbf{a}_l}{||\delta \mathbf{a}||^2} \mathbf{x}_{l-1}^\top \tag{6}$$

where $N = \sum_{l=0}^{L} N_l$ is the number of units in the network, $\delta \mathbf{a}_l = \tilde{\mathbf{a}}_l - \mathbf{a}_l$ is the activity difference between a noisy and a clean pass in layer $l$ and $\delta \mathbf{a} = (\delta \mathbf{a}_1, \ldots, \delta \mathbf{a}_L)$ is the concatenation of activity differences. The resulting update rule is referred to as activity-based node perturbation (ANP). Appendix C provides a derivation of this rule.

Here, we have now updated multiple aspects of the NP rule. First, rather than using the measure of noise injected at each layer, we instead measure the total change in activation between the clean and noisy networks. However, relative to BP our derived ANP rule has a biased estimation in gradient measurement. Appendix D describes this bias and shows that, when considering finite noise injection, this bias is measurable in closed form and is interpretable.

Second, this direct use of the activity difference also requires a recomputation of the scale of the perturbation vector. Here we carry the rescaling out by a normalization based upon the activity-difference length and then upscale the signal of this perturbation based upon the network size.

Note that the ANP rule is now a hybridization of the directional derivative and traditional node perturbation. In the limit where the nodes are not interdependent (i.e. when we have a single layer), it is equivalent to INP and upon averaging across many samples can approximate the true gradient with a distinct bias based upon correlations in propagated noise. When used in a layered network structure, it 'assumes' that all nodes are mutually independent even though they are not. This is the only remaining approximation for this approach and it is desirable to allow whole network learning without direct access to the noise vectors.

### 2.2 INCREASING NP EFFICIENCY THROUGH DECORRELATION

Uncorrelated data variables have been proposed and demonstrated as impactful in making credit assignment more efficient in deep neural networks (LeCun et al., 2002). If a layer's inputs, $\mathbf{x}_l$,

have highly correlated features, a change in one feature can be associated with a change in another correlated feature, making it difficult for the network to disentangle the contributions of each feature to the loss. This can lead to less efficient learning, as has been described in previous research in the context of BP (Luo, 2017; Wadia et al., 2021). NP additionally benefits from decorrelation of input variables at every layer. Specifically, Hiratani et al. (2022) demonstrate that the covariance of NP updates between layers $k$ and $l$ can be described as

$$C_{kl}^{\mathrm{np}} \approx 2C_{kl}^{\mathrm{sgd}} + \delta_{kl} \left\langle \sum_{m=1}^{k} \|\mathbf{g}_m\|^2 \mathbf{I}_k \otimes \mathbf{x}_{k-1} \mathbf{x}_{k-1}^T \right\rangle_{\mathbf{x}}$$

where $C_{kl}^{\mathrm{sgd}}$ is the covariance of SGD updates, $\delta_{kl}$ is the Kronecker delta and $\otimes$ is a tensor product. The above equation implies that in NP the update covariance is twice that of the SGD updates plus an additional term that depends on the correlations in the input data $\mathbf{x}_{k-1}\mathbf{x}_{k-1}^T$. Removing correlations from the input data should therefore reduce the bias in the NP algorithm updates, possibly leading to better performance.

In this work, we introduce decorrelated node perturbation, in which we decorrelate each layer's input activities using a trainable decorrelation procedure first described by Ahmad et al. (2023). A layer input $\mathbf{x}_l$ is decorrelated by multiplication by a decorrelation matrix $\mathbf{R}_l$ to yield a decorrelated input $\mathbf{x}_l^\star = \mathbf{R}_l \mathbf{x}_l$. The decorrelation matrix $\mathbf{R}_l$ is then updated according to

$$\mathbf{R}_l \leftarrow \mathbf{R}_l - \alpha \left( \mathbf{x}_l^\star \left(\mathbf{x}_l^\star\right)^\top - \mathrm{diag}\left( \left(\mathbf{x}_l^\star\right)^2 \right) \right) \mathbf{R}_l$$

where $\alpha$ is a small constant learning rate and $\mathbf{R}_l$ is initialized as the identity matrix. For a full derivation of this procedure see (Ahmad et al., 2023).

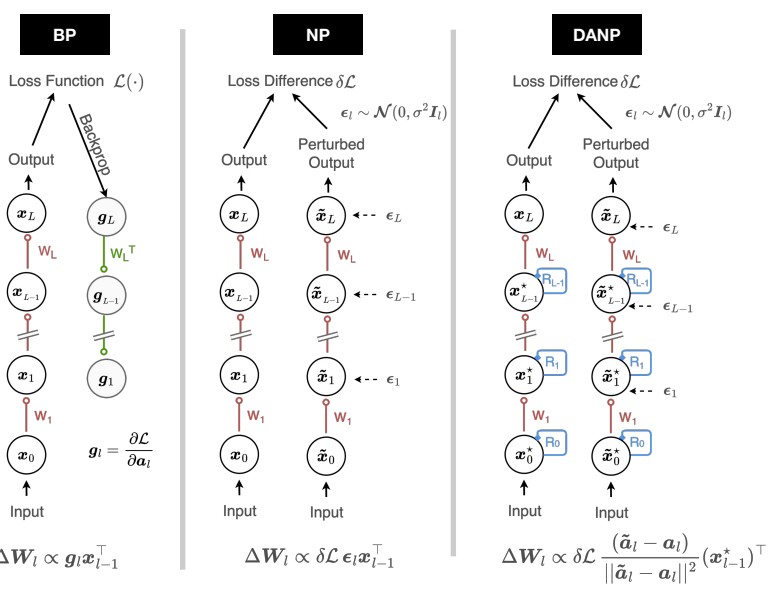

Figure 1: A graphical illustration of the computations required for update measurement for backpropagation (left), node perturbation (middle), and decorrelated activity-based node perturbation (right).

Decorrelation can be combined with any of the formulations described in the previous sections by replacing $\mathbf{x}_l$ in Eq. 1 with $\mathbf{x}_l^*$. We will use DBP, DNP, DINP and DANP when referring to the decorrelated versions of the described learning rules. Figure 1 illustrates the difference between different learning rules. In Appendix E, Algorithm 1, we describe decorrelation using the decorrelated activity-based update procedure.

## 2.3 EXPERIMENTAL VALIDATION

To measure the performance of the algorithms proposed, we ran a set of experiments with the CIFAR-10 (Krizhevsky, 2009) and Tiny ImageNet datasets (Le & Yang, 2015), using fully-connected and

convolutional neural networks, specifically aiming to quantify the performance differences between the traditional (NP), layer-wise iterative (INP) and activity-based (ANP) formulations of NP as well as their decorrelated counterparts. These datasets were chosen as they have previously been shown to yield low performance when training networks using regular node perturbation, demonstrating its limitations (Hiratani et al., 2022). All experiments were repeated using three random seeds, after which performance statistics were aggregated. Further experimental details can be found in Appendix F.

# 3 RESULTS

## 3.1 INP AND ANP ALIGN BETTER WITH THE TRUE GRADIENT

In single-layer networks, the three described NP formulations (NP, INP and ANP) converge to an equivalent learning rule. Therefore, multi-layer networks with three hidden layers were used to investigate performance differences across our proposed formulations.

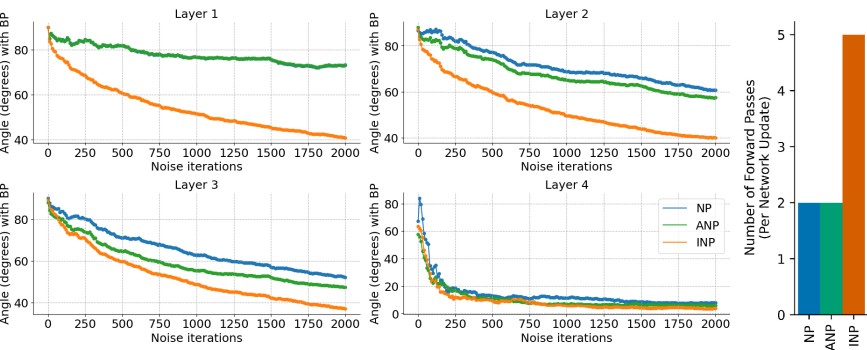

Figure 2: Left: Angles between BP's weight update and an update calculated by NP, ANP and INP as a function of the number of noise iterations (left) and the cost of each algorithm in forward passes (right). Note that the network is not updated across the 'Noise iterations' dimension, but instead multiple noise samples are propagated through a fixed network and their updates averaged. This is computed for a fully-connected, 3 hidden layer, network with leaky ReLU nonlinearities. Right: Number of forward passes for NP, ANP and INP.

Figure 2, left, shows how weight updates of the different node perturbation methods compare to those from BP in a three-hidden-layer, fully-connected, feedforward network trained on synthetic data. Specifically, it is shown how each of these methods compare in the angle of their update relative to BP when you average their update over a greater and greater number of noise samples. Note that the update for Layer 1 is identical for NP and ANP.

When measuring the angles between the update vectors of various methods, we can observe that the INP method is by far the most well-aligned in its updates with respect to backpropagation, followed by ANP and closely thereafter by NP. These results align with the theory laid out in the methods section of this work. Appendix G shows the same data as a function of the number of noisy forward passes.

Note that for all of these algorithms, alignment with BP updates improves when updates are averaged over more samples of noise. Therefore, it is the ranking of the angles between the NP algorithms that is of interest here, not the absolute value of the angle itself, as all angles would improve with more noise samples. Appendix H shows that our results do not depend strongly on the noise variance $\sigma^2$, producing similar results when varying $\sigma^2$ by a few orders of magnitude.

Figure 2, right, shows that the node perturbation methods also differ in how many forward passes are required. Specifically, although all algorithms use the same dimensionality of noise the INP method requires an individual forward pass for each layer to isolate the impact of this noise and to better approach the BP update. In the following we compare INP to the other node perturbation variants

without accounting for this excess number of forward-passes, but a reader should bear in mind this drawback and consider the added computational complexity of INP.

## 3.2 DECORRELATION IMPROVES CONVERGENCE

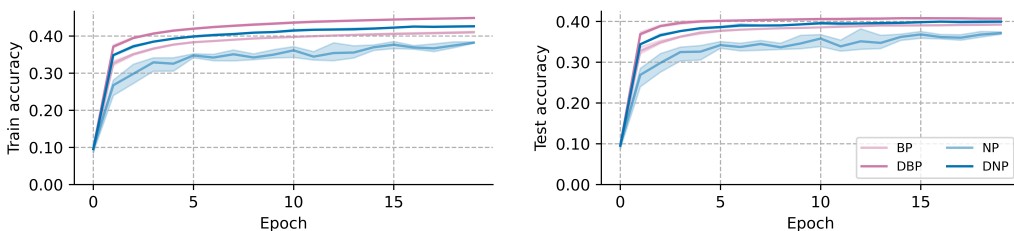

Figure 3: Performance of node perturbation and backpropagation with and without decorrelation on CIFAR-10 when training fully-connected single-layer architecture. Curves report mean train and test accuracy. Shaded areas indicate the maximal and minimal accuracy obtained for three random seeds. Note that all NP methods have an equivalent formulation in a single-layer network.

To assess the impact of decorrelation on NP's performance, we studied a single-layer network trained with NP, DNP, BP and DBP. Note that the different formulations of NP are identical in a single-layer networks where there is no impact on activities in the layer from previous layers. Figure 3 results indicate that, in a single-layer context, DNP outperforms NP significantly and performs slightly better than BP, with DBP performing better still.

It appears that part of the benefit of decorrelation is due to a lack of correlation in the input unit features and a corresponding ease in credit assignment without confound. An additional benefit from decorrelation, that is specific to NP-style updates, is explained by the way in which decorrelation reduces the covariance of NP weight updates, as described in the methods section above.

## 3.3 MULTI-LAYER NETWORKS CAN BE TRAINED EFFECTIVELY WITH NODE PERTURBATION

We proceed by determining the performance of the different algorithms in multi-layer neural networks. See Appendix I, Table 3 for peak accuracies.

Figure 4 (Top panel) shows the various network performance measures when training a multi-layer fully connected network. We see all decorrelated algorithms convincingly outperforming their non-decorrelated counterparts. We also see (D)INP outperform (D)ANP and (D)NP, almost keeping up with BP. Performance benefits of (D)ANP over (D)NP are not as clear in these results. Note that DNP performs much better in a three-hidden-layer network than in the single-layer networks explored in Figure 3, indicating that DNP does facilitate multi-layer credit assignment much more than regular NP, which actually performs worse in the deeper network. As an additional demonstration that our results generalize to other datasets Appendix J reports results on the SARCOS dataset (Vijayakumar & Schaal, 2000).

Figure 4 (Middle panel) compares the performance of DNP, DANP and DINP for 3, 6 and 9 hidden layers. Performance for DINP and DANP scales better with network depth than performance for DNP, again highlighting the improvement these novel formulations provide.

Figure 4 (Bottom panel) shows that, when training a convolutional neural network on CIFAR-10, DANP and DINP outperform NP, though relative to BP, a large gap in train accuracy emerges. Also, the NP-style algorithms lag behind BP more than they did in the fully-connected networks. All decorrelated NP variants massively outperform their non-decorrelated counterparts in both train and test accuracy. Interestingly, (D)ANP outperforms (D)NP in this experiment as well.

## 3.4 NODE PERTURBATION ALLOWS LEARNING IN NOISY SYSTEMS

One of the most interesting applications of perturbation-based methods for learning are for systems that are inherently noisy but have the property that the noise cannot be measured directly. This

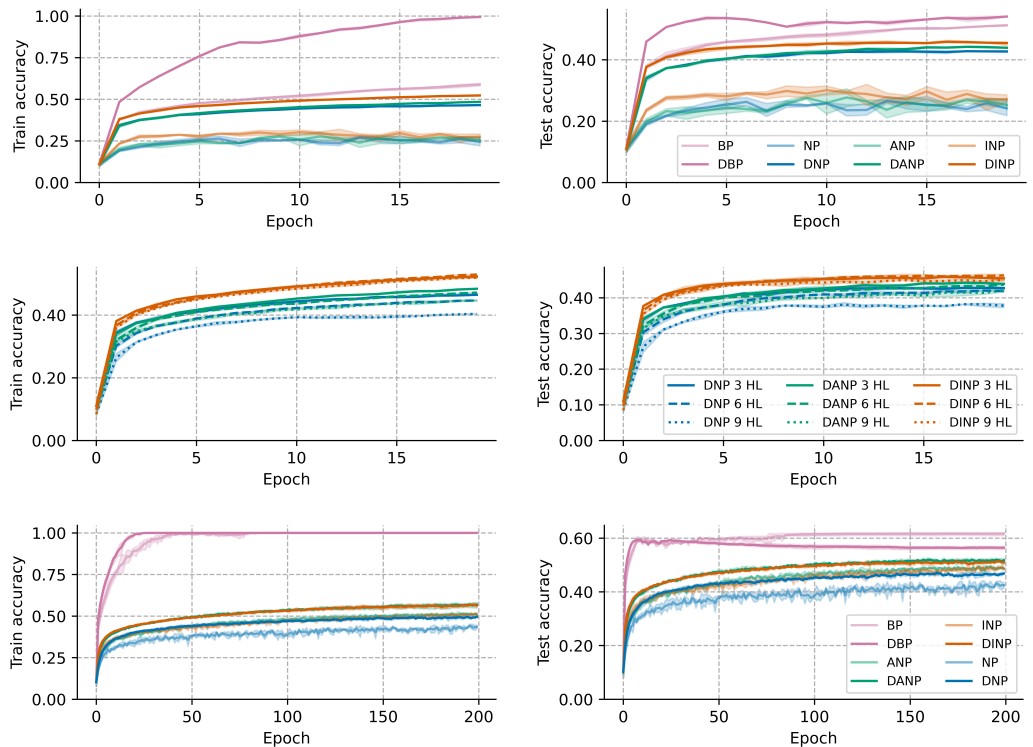

Figure 4: Performance comparisons. Curves report mean train and test accuracy. Shaded areas indicate the maximal and minimal accuracy obtained for three random seeds. Top: Performance of different NP and BP formulations on CIFAR-10 when training fully-connected three-hidden-layer architectures. Middle: Performance of DNP, DANP and DINP for 3-, 6- and 9-hidden-layer networks. Bottom: Performance of different node perturbation and backpropagation variants when training a convolutional neural network on CIFAR-10.

includes both biological nervous systems as well as a range of analog computing and neuromorphic hardware architectures (Kaspar et al., 2021).

To demonstrate that our method is also applicable to architectures with embedded noise, we train networks in which there is no clean network pass available. Instead, two noisy network passes are computed and one is taken as if it were the clean pass. That is, we use

$$\delta \mathbf{a}_l = \tilde{\mathbf{a}}_l^{(1)} - \tilde{\mathbf{a}}_l^{(2)}$$

in Eq. 6, where both $\tilde{\mathbf{a}}_l^{(1)}$ and $\tilde{\mathbf{a}}_l^{(2)}$ are generated by running a forward pass under noise perturbations. This is similar in spirit to the approach suggested by Cho et al. (2011). In this case, we specifically compare DANP to DNP in terms of their robustness to a noisy baseline. DANP does not assume that the learning algorithm can independently measure noise. Instead it can only measure the present, and potentially noisy, activity and thereafter measure activity differences to guide learning.

Figure 5 shows that computing updates based on a set of two noisy passes, rather than a clean and noisy pass, produces extremely similar learning dynamics for DANP with some minimal loss in performance. DNP, in contrast, learns more slowly during early training and becomes unstable late in training, decreasing in train accuracy. It also does not reach the same level of test performance as it does without the noisy baseline. The similarity of the speed of learning and performance levels for DANP suggests that clean network passes may provide little additional benefit for the computation of updates in this regime. These results are most promising for application of DANP to systems in which noise is an inherent property and cannot be selectively switched off or read out.

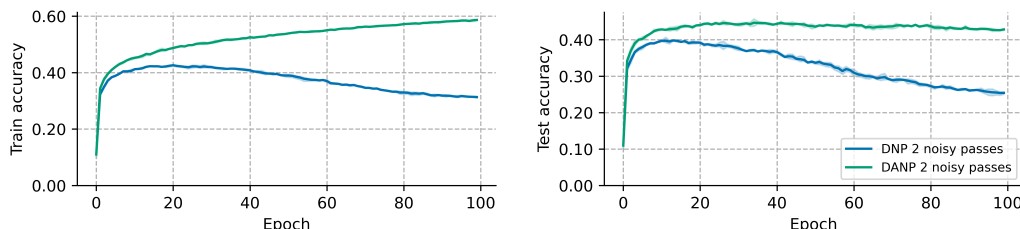

Figure 5: Performance of DNP and DANP when using two noisy network passes. Results are reported for a three-layer fully-connected network trained for CIFAR-10 classification (comparable to Figure 4, Top panel). Curves report mean train and test accuracy. Shaded areas indicate the maximal and minimal accuracy obtained for three random seeds.

### 3.5 RESAMPLING ALLOWS FOR SCALING TO LARGE SYSTEMS

To further investigate the capabilities of our approach, we apply BP, DNP, DANP and DINP to the more challenging Tiny ImageNet dataset Le & Yang (2015) using a deeper architecture consisting of three convolutional layers and three fully connected layers. The Tiny ImageNet dataset has 200 classes and thus has a larger output space than CIFAR-10. Larger output spaces are known to be challenging for traditional NP-style algorithms (Hiratani et al., 2022). RGB-values for the images, which are ImageNet images downsized to $64 \times 64$ pixels, were standardized as a preprocessing step. For network architecture and learning rates, see Appendix F.

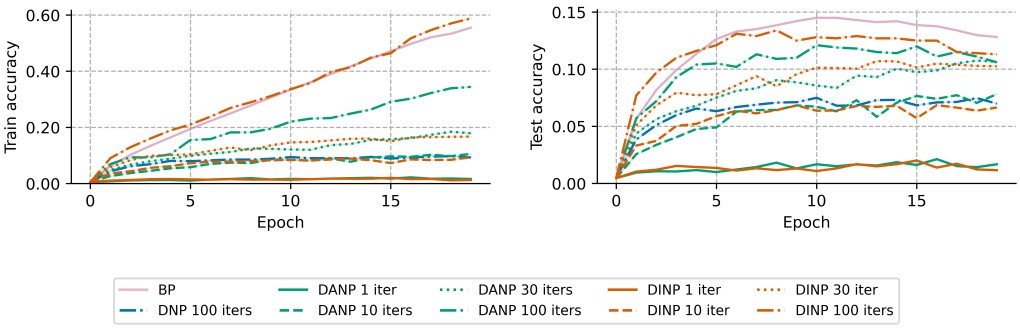

Figure 6: Train and test accuracy for BP, DNP, DANP and DINP on Tiny ImageNet. DANP and DINP applied with 1, 10, 30 and 100 noise samples per update step. DNP applied with 100 noise samples per update step.

Figure 6 shows that, in their naive implementation, DNP, DANP and DINP do not appreciably learn this task. We demonstrate that, for such a difficult task, this is due to the accuracy of the gradient estimation. To show this, we resampled the random noise 10, 30 and 100 times before applying each update, effectively carrying out 10/30/100 noisy passes to compare against each single clean pass. For DNP, only an experiment with 100 iterations was included, as this algorithm did not match the performance of DINP and DANP. Results again demonstrate that both DANP and DINP improve meaningfully on DNP in deeper networks. The additional noise sampling comes at significant computational expense but could in principle be parallelized with sufficiently powerful hardware. The results indeed show that, as our theory suggests, sampling the noise distribution multiple times per update step significantly improves alignment with backpropagation, with DINP with 100 samples per update closely approximating BP's learning trajectory. See Appendix I, Table 4 for peak accuracies.

## 4 DISCUSSION

In this work, we explored several formulations of NP learning and also introduced a layer-wise decorrelation method which strongly outperforms baseline implementations. In our results we show robust speedups in training compared to regular NP, suggesting that our alternative formulations of NP could prove to be competitive with traditional BP in certain contexts. We attribute this increased efficacy to a more efficient credit assignment by virtue of decorrelated input features at every layer, as well as an attenuation of the bias in NP updates caused by their covariance.

The performance of all of our proposed methods is substantially improved by including decorrelation. When comparing NP, INP and ANP, a number of observations are in order. First, INP yields the best results stemming from its robust relationship to the directional derivative and underlying gradient. The INP method does, however, require running as many forward passes as there are layers in the network (plus one 'clean' pass). This is a drawback for this method and should be considered when analyzing its performance, though it should also be considered that it could remain efficient on hardware in which all forward passes are parallelizable. Second, decorrelated variants of NP and ANP both perform well. An additional advantage of the latter though is that one does not need explicit access to the noise vectors, which has important implications for the development of physical learning machines, as described below.

Noise-based learning approaches might also be ideally suited for implementation on neuromorphic hardware (Kaspar et al., 2021). First, as demonstrated in Figure 5, our proposed DANP algorithm scales well even when there is no access to a 'clean' model pass. Furthermore, DANP does not require access to the noise signal itself, in contrast to NP. This means that such an approach could be ideally suited for implementation in noisy physical devices (Gokmen, 2021) even in case the noise cannot be measured. Even on traditional hardware architectures, forward passes are often easier to optimize than backward passes and are often significantly faster to compute. This can be especially true for neuromorphic computing approaches, where backward passes require automatic differentiation implementations and a separate computational pipeline for backward passes (Zenke & Neftci, 2021). In these cases, a noise-based approach to learning could prove highly efficient.

Exploring more efficient forms of noise-based learning is interesting beyond credit-assignment alone. This form of learning is more biologically plausible as it does not require weight transport nor any specific feedback processing (Grossberg, 1987; Crick, 1989; Lillicrap et al., 2016). There is ample evidence for noise in biological neural networks (Faisal et al., 2008) and we suggest here that this could be effectively used for learning. Furthermore, the positive impact of decorrelation on learning warrants further investigation of how this mechanism might be involved in neural plasticity. It is interesting to note that various mechanisms act to reduce correlation or induce whitening, especially in early visual cortical areas (King et al., 2013). Additionally, lateral inhibition, which is known to occur in the brain, can be interpreted as a way to reduce redundancy in input signals akin to decorrelation, making outputs of neurons less similar to each other (Békésy, 1967). As described in (Ahmad et al., 2023), decorrelation updates can rely exclusively on information that is locally available to the neuron, making it amenable to implementation in biological or physical systems.

Though the presented methods for effective noise-based learning show a great deal of promise, there are a number of additional research steps to be taken. The architectures considered are still relatively shallow, and thus an investigation into how well this approach scales for very deep networks would be beneficial. Testing the scalability of these approaches to tasks of greater complexity is also crucial, as are their application to other network architectures such as residual networks, recurrent networks and attention-based models. Our scaling results do demonstrate that by resampling the noise we achieve better and better alignment with the gradient direction, by virtue of the formal correspondence between directional derivatives and our formulations of node perturbation. In this sense, our approach can be viewed as an alternative to forward accumulation, where we exchange memory complexity (maintaining all partial derivatives) for time complexity (repeated forward passes with resampled noise vectors).

In general, our work opens up exciting opportunities since it has the potential to bring gradient-free training of deep neural networks within reach. That is, in addition to not requiring a backward pass, efficient noise-based learning may also lend itself to networks not easily trained by backpropagation, such as those consisting of activation functions with jumps, binary networks or networks in which the computational graph is broken, as in reinforcement learning.

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

## A  WEIGHTS PROPAGATE NOISE CORRELATIONS

Traditional NP updates have a number of sources of error. One reason is that use of the injected noise $\epsilon_l$ to compute the learning signal ignores how the output activity of layer $l$ is not only impacted by the noise injected directly into it, but also by the cumulative effect of perturbations added to previous layers. If the noise is randomly sampled from a Gaussian distribution with mean zero, one might be tempted to simply assume that the perturbations from previous layers cancel out in expectation, but this assumption ignores the correlations introduced into these perturbations by the network's weight matrices at all preceding layers. Multiplying a random vector by a non-orthogonal matrix will introduce a non-random covariance structure into it. To see why, consider a transformation $\mathbf{Wx} = \mathbf{y}$, where $\mathbf{W}$ is a randomly initialized weight matrix and $\mathbf{x}$ is an uncorrelated input vector for which it holds that $\langle \mathbf{xx}^\top \rangle = \mathbf{I}$ with $\mathbf{I}$ the identity matrix. The covariance matrix of $\mathbf{y}$ can be described as

$$\text{cov}_\mathbf{y} = \langle \mathbf{yy}^\top \rangle = \langle (\mathbf{Wx})(\mathbf{Wx})^T \rangle = \langle (\mathbf{Wx})(\mathbf{x}^\top \mathbf{W}^\top) \rangle$$
$$= \langle \mathbf{Wxx}^\top \mathbf{W}^\top \rangle = \mathbf{W} \langle \mathbf{xx}^\top \rangle \mathbf{W}^\top = \mathbf{WIW}^\top = \mathbf{WW}^\top$$

Therefore, any matrix for which $\mathbf{WW}^\top \neq \mathbf{I}$ will add some covariance structure into output vector $\mathbf{y}$, even when input vector $\mathbf{x}$ is uncorrelated.

## B  THE DIRECTIONAL DERIVATIVE AS A MEASURE OF THE GRADIENT

In the main text, we relate the measurement of directional derivatives to the gradient. Specifically, for a feedforward deep neural network of $L$ layers, we state that

$$\mathbf{g}_l = N_l \left\langle \nabla_\mathbf{v} \mathcal{L} \frac{\mathbf{v}_l}{||\mathbf{v}||} \right\rangle_\mathbf{v}$$

where $\mathbf{v} = (\mathbf{v}_1, \ldots, \mathbf{v}_L)$ and $\mathbf{v}_k \sim \mathcal{N}(\mathbf{0}, \sigma^2 \mathbf{I}_k)$ if $k = l$ and $\mathbf{v}_k = \mathbf{0}$ otherwise.

Let us now demonstrate the equivalence of the gradient to the expectation over this directional derivative. For simplicity, let us consider a single layer network, such that $L = 1$ and $\mathbf{v} = \mathbf{v}_l = \mathbf{v}_1$. A directional derivative can always be equivalently written as the gradient vector dot product with a (unit-length) direction vector, such that

$$\nabla_\mathbf{v} \mathcal{L} = \nabla \mathcal{L}^\top \frac{\mathbf{v}}{||\mathbf{v}||} \,.$$

Substituting this form into the above equation requires ensuring that our directional derivative term is no longer treated as a scalar but as a $\mathbb{R}^{1 \times 1}$ matrix. We also a transpose the gradient vector and then untranspose the entire expression to allow derivation with clarity. Further, we remove the now redundant subscripts, such that:

$$\mathbf{g} = N \left\langle \nabla_\mathbf{v} \mathcal{L} \frac{\mathbf{v}^\top}{||\mathbf{v}||} \right\rangle_\mathbf{v}^\top = N \left\langle \nabla \mathcal{L}^\top \frac{\mathbf{v}}{||\mathbf{v}||} \frac{\mathbf{v}^\top}{||\mathbf{v}||} \right\rangle_\mathbf{v}^\top = N \left\langle \frac{\mathbf{vv}^\top}{||\mathbf{v}||^2} \right\rangle_\mathbf{v}^\top \nabla \mathcal{L} \,.$$

If we assume that our noise distribution is composed of independent noise with a given variance, $\boldsymbol{\Sigma} = \sigma^2 \mathbf{I}$ then we obtain

$$\mathbf{g} = N \left\langle \frac{\mathbf{vv}^\top}{||\mathbf{v}||^2} \right\rangle_\mathbf{v}^\top \nabla \mathcal{L} \propto N \frac{\mathbf{I}}{N} \nabla \mathcal{L} = \nabla \mathcal{L} \,.$$

Note that normalization of every vector means that the correlations $\left\langle \mathbf{vv}^\top / ||\mathbf{v}||^2 \right\rangle$ are not changed in sign but only in scale, as the noise vectors are now all scaled to lie on a unit sphere (without any rotation). In our specific case of a diagonal correlation matrix, this expectation is also therefore diagonal and all diagonal elements are equal. The average value of these diagonal elements is then simply the variance $1/N$ of a random unit vector in $N$ dimensional space. For other noise distributions, in which there does exist cross-correlation between the noise elements, this is no longer an appropriate treatment. Instead, one would have to multiply by the inverse of the correlation matrix to fully recover the gradient values.

## C  DERIVATION OF ACTIVITY-BASED NODE PERTURBATION

Here we explicitly demonstrate how the INP rule can give rise to the ANP rule under some limited assumptions. Consider the INP learning rule, which is designed to return a weight update for a specific layer of a DNN,

$$\Delta\mathbf{W}_l^{\text{INP}} = N_l \, \delta\mathcal{L} \frac{\mathbf{v}_l}{||\mathbf{v}||^2} \, \mathbf{x}_{l-1}^\top \, .$$

This rule has been derived in the main text in order to optimally make use of noise to determine the directional derivative with respect to a single layer of a deep network, $l$, while all other layers receive no noise. For this purpose, the noise vector is defined such that $\mathbf{v}_k \sim \mathcal{N}(\mathbf{0}, \sigma^2 \mathbf{I}_k)$ if $k = l$ and $\mathbf{v}_k = \mathbf{0}$ otherwise. In moving from INP to ANP, we aim to fulfill a number of conditions:

1. Update a whole network in one pass rather than per layer.

2. Allow updating without explicit access to the noise at every layer, but instead access only to the activity difference.

These goals can be accomplished in a simple manner. To achieve the first goal, we can simply treat the whole network as if it is a single layer (even if it is not). The only change required for this modification is to assume that this rule holds even if $\mathbf{v}_k \sim \mathcal{N}(\mathbf{0}, \sigma^2 \mathbf{I}_k)$ for $k \in [1, \ldots, L]$. I.e., that noise is injected for all layers and all layers are simultaneously updated. To achieve the second goal, we can assume that one does not have access to the noise vector directly, but instead has access to the output activations from a clean and noisy pass, $\mathbf{a}_l$ and $\tilde{\mathbf{a}}_l$ respectively. Thus, rather than measuring the noise directly, one can measure the impact of all noise upon the network by substituting the noise vector, $\mathbf{v}$ for the activity difference $\delta\mathbf{a} = \tilde{\mathbf{a}} - \mathbf{a}$. Thus, with these two modifications we arrive at the ANP learning rule

$$\Delta\mathbf{W}_l^{\text{ANP}} = N \, \delta\mathcal{L} \frac{\delta\mathbf{a}_l}{||\delta\mathbf{a}||^2} \, \mathbf{x}_{l-1}^\top$$

which can be computed via two forward passes only.

## D  BIAS IN NP COMPARED TO ANP

In this section, we analyze the deviation of NP's and ANP's updates from the true gradient in a simple two-layer neural network, focusing on the effects of noise propagation through the network's layers. Unlike existing work in the literature, we do not assume infinitely small perturbations, or a linear network, to provide a quantification of bias in NP's and ANP's updates. In practice, perturbations applied to networks cannot be infinitely small and are even limited by precision of floating point representations to be larger than some minimum size.

Consider a two-layer neural network with weight matrices $\mathbf{W}_1$ and $\mathbf{W}_2$, no biases, and a nonlinear activation function $f(\cdot)$ applied after the first layer. The network's output, given an input $x$, is expressed as:

$$\mathbf{y}_2 = \mathbf{W}_2 f(\mathbf{W}_1 \mathbf{x}) \, .$$

Noise is introduced into the network as small perturbations $\mathbf{v}_1$ and $\mathbf{v}_2$ at the outputs of the first and second layers, respectively. With this perturbation, the perturbed output becomes:

$$\tilde{\mathbf{y}}_2 = \mathbf{W}_2 \big( f(\mathbf{W}_1 \mathbf{x} + \mathbf{v}_1) + \mathbf{v}_2 \big) \, .$$

Given an arbitrary loss function $L$, the loss differential is given by $L(\tilde{\mathbf{y}}_2) - L(\mathbf{y}_2)$. NP's updates are based on a product between this loss differential and the noise. Examining this for matrix $\mathbf{W}_2$, this corresponds to a product between the noise differential and vector $\mathbf{v}_2$. To fully show the impact of this update, we expand the loss differential by a Taylor expansion beyond the first term (i.e. assuming that our loss difference is not infinitely small or linearizable) and then further expand the activity

difference by Taylor expansion. As such,

$$
\langle \Delta \mathbf{W}_2^{\mathrm{NP}} \rangle = \frac{1}{h} \langle (L(\tilde{\mathbf{y}}_2) - L(\mathbf{y}_2)) \cdot \mathbf{v}_2 \rangle f(\mathbf{y}_1)^\top
$$

$$
= \frac{1}{h} \left\langle \mathbf{v}_2 \left[ \nabla_{\mathbf{y}_2} L^\top (\tilde{\mathbf{y}}_2 - \mathbf{y}_2) + \frac{1}{2!} (\tilde{\mathbf{y}}_2 - \mathbf{y}_2)^\top \nabla_{\mathbf{y}_2}^2 L\ (\tilde{\mathbf{y}}_2 - \mathbf{y}_2) + \frac{1}{3!} \cdots \right]^\top \right\rangle f(\mathbf{y}_1)^\top
$$

$$
= \frac{1}{h} \left\langle \mathbf{v}_2 \left[ \nabla_{\mathbf{y}_2} L^\top \left( \mathbf{v}_2 + \nabla_{\mathbf{y}_1} \mathbf{y}_2^\top \mathbf{v}_1 + \frac{1}{2!} \mathbf{v}_1^\top \nabla_{\mathbf{y}_1}^2 \mathbf{y}_2 \mathbf{v}_1 + \frac{1}{3!} \cdots \right) \cdots \right]^\top \right\rangle f(\mathbf{y}_1)^\top
$$

$$
= \frac{1}{h} \left\langle \mathbf{v}_2 \left[ \left( \mathbf{v}_2 + \nabla_{\mathbf{y}_1} \mathbf{y}_2^\top \mathbf{v}_1 + \frac{1}{2!} \mathbf{v}_1^\top \nabla_{\mathbf{y}_1}^2 \mathbf{y}_2 \mathbf{v}_1 + \frac{1}{3!} \cdots \right)^\top \nabla_{\mathbf{y}_2} L \cdots \right] \right\rangle f(\mathbf{y}_1)^\top
$$

$$
= \frac{1}{h} \left\langle \mathbf{v}_2 \mathbf{v}_2^\top + \mathbf{v}_2 \mathbf{v}_1^\top \nabla_{\mathbf{y}_1} \mathbf{y}_2 + \frac{1}{2!} \mathbf{v}_2 \mathbf{v}_1^\top \nabla_{\mathbf{y}_1}^2 \mathbf{y}_2^\top \mathbf{v}_1 + \frac{1}{3!} \cdots \right\rangle \nabla_{\mathbf{y}_2} L\ f(\mathbf{y}_1)^\top
$$

$$
= \cdots
$$

The above Taylor expansion demonstrates that above the linear terms $(\mathbf{v}_2 \mathbf{v}_2^\top + \mathbf{v}_2 \mathbf{v}_1^\top \nabla_{\mathbf{y}_1} \mathbf{y}_2 \approx \mathbf{I} + \mathbf{0}$, in the ideal noise case), there are terms involving the correlation between noise vector $\mathbf{v}_2$ and higher order terms involving $\mathbf{v}_1$, beginning with the Hessian but also including third, fourth and higher derivative terms. We cannot, in general, say anything about the unbiasedness of these terms and these are sources of unexplained error in existing work.

By comparison, ANP's Taylor expansion has a much more interpretable correlation structure. Again looking at weight updates for layer two:

$$
\langle \Delta \mathbf{W}_2^{\mathrm{ANP}} \rangle = \frac{1}{h} \left\langle (L(\tilde{\mathbf{y}}_2) - L(\mathbf{y})) \cdot (\tilde{\mathbf{y}}_2 - \mathbf{y}_2) \right\rangle f(\mathbf{y}_1)^\top
$$

$$
= \frac{1}{h} \left\langle (\tilde{\mathbf{y}}_2 - \mathbf{y}_2) \left[ \nabla_{\mathbf{y}_2} L^\top (\tilde{\mathbf{y}}_2 - \mathbf{y}_2) + \frac{1}{2!} (\tilde{\mathbf{y}}_2 - \mathbf{y}_2)^\top \nabla_{\mathbf{y}_2}^2 L (\tilde{\mathbf{y}}_2 - \mathbf{y}_2) + \frac{1}{3!} \cdots \right]^\top \right\rangle f(\mathbf{y}_1)^\top
$$

$$
= \frac{1}{h} \left\langle (\tilde{\mathbf{y}}_2 - \mathbf{y}_2)(\tilde{\mathbf{y}}_2 - \mathbf{y}_2)^\top \nabla_{\mathbf{y}_2} L + \frac{1}{2!} (\tilde{\mathbf{y}}_2 - \mathbf{y}_2)(\tilde{\mathbf{y}}_2 - \mathbf{y}_2)^\top \nabla_{\mathbf{y}_2}^2 L^\top (\tilde{\mathbf{y}}_2 - \mathbf{y}_2) + \frac{1}{3!} \cdots \right]^\top \right\rangle f(\mathbf{y}_1)^\top .
$$

This simplifies all assumptions regarding higher-order noise propagation terms, and reduces the algorithms bias down to the correlation in overall activity differences $(\langle (\tilde{\mathbf{y}}_2 - \mathbf{y}_2)(\tilde{\mathbf{y}}_2 - \mathbf{y}_2)^\top \rangle)$. This is a much more interpretable bias term which could even be targeted directly to be made diagonal if a practitioner was interested in doing so.

# E  ALGORITHM PSEUDOCODE

---

**Algorithm 1** Decorrelated activity-based node perturbation (DANP)

---

**Input:** *data* $\mathcal{D}$, *network* $\{(\mathbf{W}_l, \mathbf{R}_l)\}_{l=1}^L$, *learning rates* $\eta$ and $\alpha$
**for each** *epoch* **do**
    **for each** $(\mathbf{x}_0, \mathbf{t}) \in \mathcal{D}$ **do**
        **for** layer $l$ **from** 1 **to** $L$ **do**                            ▷ Regular forward pass
           $\mathbf{x}_{l-1}^\star = \mathbf{R}_{l-1} \mathbf{x}_{l-1}$
           $\mathbf{a}_l = \mathbf{W}_l \mathbf{x}_{l-1}^\star$
           $\mathbf{x}_l = f(\mathbf{a}_l)$
        **end for**
        $\tilde{\mathbf{x}}_0 = \mathbf{x}_0$
        **for** layer $l$ **from** 1 **to** $L$ **do**                            ▷ Noisy forward pass
           $\tilde{\mathbf{x}}_{l-1}^\star = \mathbf{R}_{l-1} \tilde{\mathbf{x}}_{l-1}$
           $\tilde{\mathbf{a}}_l = \mathbf{W}_l \tilde{\mathbf{x}}_{l-1}^\star + \boldsymbol{\epsilon}_l$
           $\tilde{\mathbf{x}}_l = f(\tilde{\mathbf{a}}_l)$
        **end for**
        $\delta\mathcal{L} = (\mathbf{t} - \tilde{\mathbf{x}}_L)^2 - (\mathbf{t} - \mathbf{x}_L)^2$                  ▷ Compute loss difference
        **for** layer $l$ **from** 1 **to** $L$ **do**
           $\mathbf{W}_l \leftarrow \mathbf{W}_l - \eta N\, \delta\mathcal{L}\, \frac{\tilde{\mathbf{a}}_l - \mathbf{a}_l}{||\delta\mathbf{a}||^2} \left(\mathbf{x}_{l-1}^\star\right)^\top$           ▷ Update weight matrix
           $\mathbf{R}_l \leftarrow \mathbf{R}_l - \alpha \left( \mathbf{x}_l^\star \left(\mathbf{x}_l^\star\right)^\top - \mathrm{diag}\left( (\mathbf{x}_l^\star)^2 \right) \right) \mathbf{R}_l$      ▷ Update decorrelation matrix
        **end for**
    **end for**
**end for**

---

# F    EXPERIMENTAL DETAILS

Experiments were performed using fully-connected or convolutional neural networks with leaky ReLU activation functions and a categorical cross-entropy (CCE) loss on the one-hot encoding of class membership. The Adam optimizer with default parameters $beta_1 = 0.9$, $beta_2 = 0.999$, $\epsilon = 1e - 7$, was used for optimization. All experiments were run on an HP OMEN GT13-0695nd desktop computer with an NVIDIA RTX 3090 GPU. The experiments required about a week of total computation time and several more weeks of computation time during exploratory experiments.

Table 1 provides details of the employed neural network architectures.

Table 1: The neural network architectures consist of fully connected (FC) and convolutional (Conv) layers. All layers except the output layer are followed by a leaky ReLU transformation. Convolutional layer size is given as height$\times$width$\times$output channels, stride.

| NETWORK | LAYER TYPES | LAYER SIZE |
|---|---|---|
| SINGLE LAYER | FC | 10 |
| THREE HIDDEN LAYERS | $3 \times$ FC | 1024 |
|  | FC | 10 |
| SIX HIDDEN LAYERS | $6 \times$ FC | 1024 |
|  | FC | 10 |
| NINE HIDDEN LAYERS | $9 \times$ FC | 1024 |
|  | FC | 10 |
| CONVNET | CONV | $3\times3\times16$, 2 |
|  | CONV | $3\times3\times32$, 2 |
|  | CONV | $3\times3\times64$, 1 |
|  | FC | 1024 |
|  | FC | 10 |
| CONVNET TINY IMAGENET | CONV | $5\times5\times32$, 2 |
|  | CONV | $3\times3\times64$, 2 |
|  | CONV | $5\times5\times128$, 1 |
|  | FC | 512 |
|  | FC | 512 |
|  | FC | 200 |

Table 2 shows the learning rates $\eta$ used in each experiment. For each experiment, the highest stable learning rate was selected. For the decorrelation learning rate a fixed value of $\alpha = 10^{-3}$ was chosen based on a prior manual exploration. Note that the minibatch size was fixed for every method at 1000. When using NP methods, activity perturbations were drawn from a univariate Gaussian with variance $\sigma^2 = 10^{-6}$.

Table 2: Learning rates used for different learning algorithms and architectures.

| METHOD | 1 LAYER | 3 HIDDEN LAYERS | 6 HIDDEN LAYERS | 9 HIDDEN LAYERS | CONVNET | CONVNET TINY IMAGENET |
|---|---|---|---|---|---|---|
| NP | $1.0 \times 10^{-3}$ | $1.0 \times 10^{-4}$ | – | – | – | – |
| DNP | $1.0 \times 10^{-3}$ | $1.0 \times 10^{-3}$ | $5.0 \times 10^{-3}$ | $5.0 \times 10^{-3}$ | – | $1.0 \times 10^{-3}$ |
| ANP | – | $1.0 \times 10^{-4}$ | – | – | $1.0 \times 10^{-4}$ | – |
| DANP | – | $1.0 \times 10^{-3}$ | $5.0 \times 10^{-3}$ | $5.0 \times 10^{-3}$ | $1.0 \times 10^{-4}$ | $1.0 \times 10^{-3}$ |
| INP | – | $1.0 \times 10^{-4}$ | – | – | $1.0 \times 10^{-4}$ | – |
| DINP | – | $1.0 \times 10^{-3}$ | $5.0 \times 10^{-3}$ | $5.0 \times 10^{-3}$ | $1.0 \times 10^{-4}$ | $1.0 \times 10^{-3}$ |
| BP | $1.0 \times 10^{-3}$ | $1.0 \times 10^{-4}$ | – | – | $1.0 \times 10^{-3}$ | $1.0 \times 10^{-4}$ |
| DBP | $1.0 \times 10^{-3}$ | $1.0 \times 10^{-3}$ | – | – | $1.0 \times 10^{-3}$ | – |

## G  ALIGNMENT AS A FUNCTION OF NOISY FORWARD PASSES

Figure 7 shows the same data as Figure 2, but the x-axis displays the number of noisy forward passes performed, instead of the number of noise iterations. This figure essentially compares NP, ANP and INP given the same amount of computation.

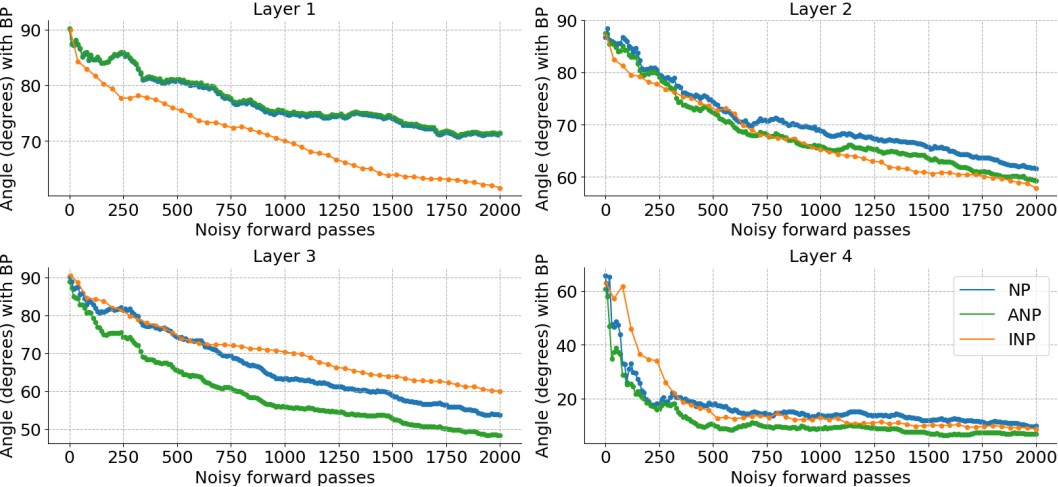

Figure 7: Angles between BP's weight update and an update calculated by NP, ANP and INP as a function of the number of noisy forward passes.

## H  IMPACT OF PERTURBATION SCALE ON PERFORMANCE

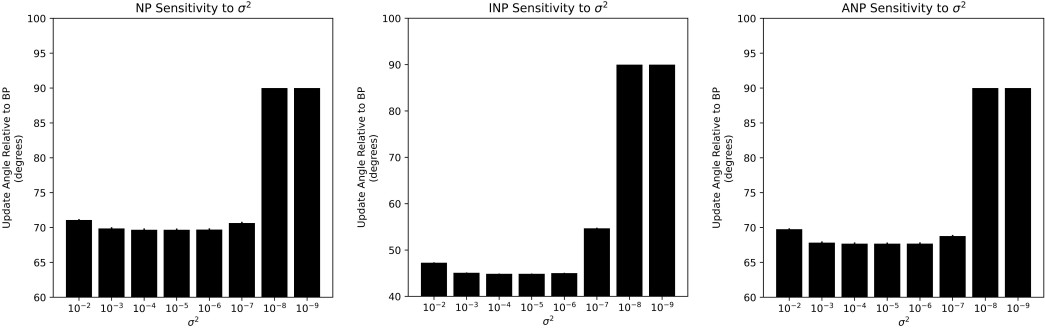

Figure 8: The alignment of NP, INP, and ANP updates measured against BP are shown across a range of perturbation parameters ($\sigma^2$). These angles are measure from updates computed for the second hidden layer of a randomly initialized three-hidden layer network being using a 1000 sample mini-batch from the CIFAR-10 dataset. The network from which these are sampled is equivalent to that used in Figure 2.

In all simulations of this paper, random noise is drawn from a set of independent Gaussian distributions with zero mean and some variance, $\sigma^2$. As can be seen in Figure 8, the NP, INP, and ANP methods show a robust performance (when measured based upon alignment with BP updates) to the choice of this variance of the perturbation distribution. For values across multiple orders of magnitude, from $10^{-4}$ to $10^{-6}$, there performance is completely stable. For smaller values of the variance, $\sigma^2 < 10^{-7}$, precision errors cause a decrease in performance (increase in angle). And for larger values of the variance, $\sigma^2 > 10^{-3}$, the size of the perturbation begins to affect network dynamics and thus the

approximation of the directional derivative. In this work we exclusively use $\sigma^2 = 10^{-6}$, the smallest value possible before precision errors become an issue. We do not find significantly different results, even if $\sigma^2$ is set to be an order of magnitude greater.

## I  PEAK ACCURACIES

Table 3 shows peak accuracies for all algorithms used in the CIFAR-10 experiments and Table 4 shows peak accuracies for the Tiny ImageNet experiment.

Table 3: Peak percentage accuracies for different learning algorithms and architectures on CIFAR-10. Best performance across all methods is shown in boldface.

| METHOD | 1 LAYER | | 3 HIDDEN LAYERS | | 6 HIDDEN LAYERS | | 9 HIDDEN LAYERS | | CONVNET | |
| | TRAIN | TEST | TRAIN | TEST | TRAIN | TEST | TRAIN | TEST | TRAIN | TEST |
| --- | --- | --- | --- | --- | --- | --- | --- | --- | --- | --- |
| BP | 41.3 | 39.3 | 59.6 | 51.5 | – | – | – | – | **100.0** | **62.1** |
| DBP | **44.9** | **40.9** | **99.6** | **54.3** | – | – | – | – | **100.0** | 60.5 |
| NP | 38.5 | 38.2 | 28.1 | 28.1 | – | – | – | – | 45.2 | 44.5 |
| DNP | 42.8 | 40.2 | 44.9 | 42.1 | 46.7 | 43.1 | 40.5 | 38.6 | 50.2 | 47.9 |
| ANP | – | – | 30.4 | 30.6 | – | – | – | – | 52.7 | 49.9 |
| DANP | – | – | 48.5 | 44.4 | 47.3 | 44.2 | 44.8 | 42.7 | 58.2 | 52.6 |
| INP | – | – | 32.0 | 32.0 | – | – | – | – | 52.3 | 49.9 |
| DINP | – | – | 52.6 | 46.3 | **53.0** | **46.6** | **52.0** | **45.3** | 57.9 | 52.2 |

Table 4: Peak percentage accuracies for different learning algorithms on Tiny ImageNet. Best performance across all methods is shown in boldface.

| METHOD | TINY IMAGENET | |
| | TRAIN | TEST |
| --- | --- | --- |
| BP | 55.5 | **14.5** |
| DANP 1 ITER | 2.2 | 2.1 |
| DANP 10 ITERS | 13.5 | 9.0 |
| DANP 30 ITERS | 32.8 | 11.3 |
| DANP 100 ITERS | 35.4 | 12.0 |
| DINP 1 ITER | 2.0 | 2.0 |
| DINP 10 ITERS | 11.4 | 8.1 |
| DINP 30 ITERS | 26.7 | 11.8 |
| DINP 100 ITERS | **60.8** | 13.4 |
| DNP 100 ITERS | 9.7 | 7.5 |

# J  SARCOS EXPERIMENT

Figure 9 shows train and test loss for (D)NP, (D)ANP and (D)BP for the SARCOS dataset (Vijayaku-mar & Schaal, 2000). The task is to solve inverse dynamics problem for a seven degrees-of-freedom SARCOS anthropomorphic robot arm by mapping from a 21-dimensional input space (7 joint positions, 7 joint velocities, 7 joint accelerations) to the corresponding 7 joint torques. The network is a fully connected architecture with two 32-unit hidden layers and was trained with MSE loss and the leaky ReLU activation function. The network was trained for 500 epochs and results were averaged over three random seeds. A learning rate search for each algorithm was started at $10^{-7}$, doubling the learning rate until the algorithm became unstable. A decorrelation learning rate of $10^{-4}$ was used for DNP and DANP and $10^{-3}$ for DBP.

The results indicate that while NP and ANP lag BP in performance, DNP and DANP perform almost as well as BP. Like in other experiments reported in this work, DBP performs best overall. Final test losses were 11.18 (NP), 10.51 (DNP), 15.86 (ANP), 9.79 (DANP), 9.48 (BP) and 7.02 (DBP).

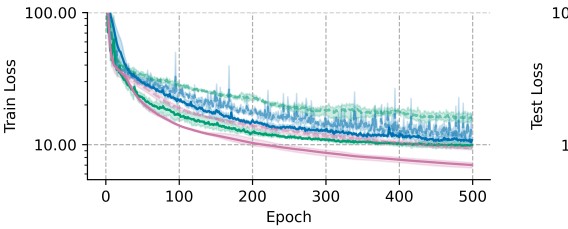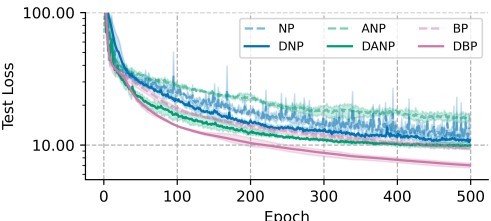

Figure 9: Train and test loss for (D)NP, (D)ANP and (D)BP on the SARCOS dataset. Curves report mean train and test loss. Error bars indicate the maximal and minimal loss obtained for three random seeds.

