# OpenReview forum: "Effective Learning with Node Perturbation in Multi-Layer Neural Networks"
_ICLR.cc/2025/Conference — Submitted to ICLR 2025_

### Official Review · Reviewer_wuaz · 2024-10-28

**Soundness:** 3
**Presentation:** 3
**Contribution:** 3
**Rating:** 6
**Confidence:** 4

**Summary:**

In this paper, the authors extend an existing framework to train multi-layer neural networks without using backpropagation (BP). Node perturbation (NP) relies on injecting noise in the network layers, measuring the in the loss differential between the clean and noisy pass, and computing a layer-wise update which relies on the noise vector, the pre-synaptic activity and the loss differential. The authors improve on the NP framework with three main contributions:
* In the traditional NP approach, the effect of the layer $\ell$ noise $\epsilon_\ell$ on the downstream layers is unaccounted for. By computing the directional derivative of the loss on the noise injected at layer $\ell$ ($\nabla_v \mathcal{L}$), they can more precisely target the updates to layer $\ell$. This method, referred to as iterative node perturbation, requires $L+1$ forward passes, and relies on access to the noise vector.
* Next, the authors propose activity-based noise perturbation (ANP). This approach relies on the assumption that all the layers are independent, which requires measuring the state difference between clean and noisy passes instead of the noise injected. This requires only two passes and does not require access to the noise signal, but rather its effect on the network.
* Lastly, using an existing trainable decorrelation procedure, they show improved performance of their proposed algorithms by decorrelating the inputs to each layer.

These variations of NP are tested fully connected and convolutional networks on CIFAR-10 and Tiny ImageNet and compared to BP.

**Strengths:**

I believe the paper is well structured and well written, the results are interesting and overall well defended. In detail:
* The methods section is clear and straightforward. I appreciated the incremental structure that starts from Node Perturbation ad adds on the newly proposed variations explaining well the contribution of each piece.
* The results prove the claims of the authors regarding how each variation of NP compares, and I appreciate using Tiny ImageNet as a benchmark which is more challenging of what is usually found in these types of papers.
* I found section 3.4 very interesting, as most methods I know of rely on having non-noisy systems. I think removing the assumption of having a clean and noisy pass, and assuming all passes are noisy makes it a very interesting algorithm.

**Weaknesses:**

I am giving this paper a 6 because of the following weaknesses I found, but I would be happy to re-evaluate if these concerns are addressed. My main concerns are:
* The world bioplausible is mentioned throughout the papers, but details on how these algorithms could be implemented in biological neural networks are not provided. I believe the paper still stands without needing to justify it as bioplausible, so I believe that either removing the bioplausibility aspect (and exchange it with more details on possible hardware implementation) or providing more details about the bioplausibility would be better alternatives.
* In the last years, many alternatives to back propagation that rely on multiple forward passes have been proposed. In particular, looking at equation (6) in the paper. For example, Dellaferrera and Kreiman, 2022, proposes to use the differential between a clean and "noisy pass" to train the network, where the noisy pass relies on a perturbation of the activity computed using the loss differential. Since there are many other algorithms like this (for example Hinton 2022, Kohan et al. 2018, Frenkel et al 2021), I believe a comparison in the methods section would be beneficial.
* The figures position could be improved to make the paper more readable. For example Figure 1 could be pushed up in the paper as it summarizes the contributions of the paper, Figure 4 could be moved in section 3.3.
* I think figure 2 could be improved. For example figures 2 right does not add anything that cannot be explained in words.

**Questions:**

These questions/suggestions mostly add on top of what I mentioned in the weaknesses:
* What happens to the update angle during the training. Do you observe a better alignment close to convergence? If this is interesting it could be added to Figure 2.
* I am pretty surprised about the results with a fully connected network trained with BP, as it is pretty common to see these networks obtaining >60% accuracy on CIFAR-10. Could you elaborate more on the choice of hyperparameters?
* I believe is always important to include a code repository in these papers as it helps the other researchers in the field and makes the experiments easier to reproduce.

---

> ### Author Response · Authors · 2024-11-26
> **Response to the reviewer's comments**
>
> We thank the reviewer for their careful and constructive feedback.
>
> __Regarding the details of bioplausibility__
> Our aim in this work was not to implement a complete biologically ‘detailed’  algorithm per se, but to study a gradient-free learning method with several possible applications and implementations. DANP, in particular, is able to learn in systems which do not have a noise-free baseline, based upon a global feedback signal, which is more akin to how biological brains learn and is also applicable to certain types of hardware.
>
> __Regarding other relevant work__
> Though our paper specifically investigates noise-based algorithms, a brief mention of other bio-plausible algorithms has now been added to the introduction, to contrast them with these noise-based algorithms.
>
> __Regarding figure layout and the usefulness of Figure 2__
> Figure 2 has been replaced by a more in-depth figure, exploring the alignment of NP, ANP and INP with BP as a function of the number of noisy forward passes. For the camera ready version we will have a deeper look at layout issues.
>
> __Regarding update angle during training__
> We have added a new experiment studying the update angle as a function number of noise iterations again in Figure 2 to shed some more light on the update angles of the various algorithms.
>
> __Regarding performance levels on CIFAR__
> Performance in all experiments slightly lagging behind established benchmarks is likely due to the simple networks used. The aim of this work was to compare the convergence properties of algorithms on a decently challenging task, not to achieve any set level of benchmark performance. We selected learning rates for each experiment separately based on a grid search, see Appendix F. Other hyper-parameters, like noise magnitude, did not seem to affect performance much as long as they stayed within a reasonable range. This is described in Appendix H.
>
> __Regarding code repository__
> An implementation of the algorithms will be made available upon acceptance.

---

> > ### Comment · Reviewer_wuaz · 2024-11-26
> >
> > I want to thank the reviewers for addressing my concerns. I read the revised version of the paper and I believe that most of my concerns have been addressed. Regarding the BP benchmark, I have read a number of papers which achieve ~60% accuracy on CIFAR-10 on 1-hidden layers network. Although I do not think this is extremely important for the paper itself, I just want to make sure the comparison is done rigorously.
> >
> > Although I will wait to read the discussion with the other reviewers to update my score, as of now, I would be happy to recommend this paper for acceptance.

---

### Official Review · Reviewer_smGL · 2024-11-03

**Soundness:** 3
**Presentation:** 3
**Contribution:** 3
**Rating:** 6
**Confidence:** 4

**Summary:**

Backpropagation (BP) is the standard for training deep neural networks but is criticized for its lack of biological plausibility and computational complexity due to separate forward and backward phases. Node Perturbation (NP) offers an alternative approach by injecting noise into hidden layer activities and using the resulting change in the loss function to guide weight updates. However, traditional NP is inefficient, unstable, and requires precise noise control, limiting its practical utility and biological relevance.

This study extends NP by introducing more robust formulations. It reframes NP using the concept of directional derivatives, leading to an iterative approach (INP) that better aligns with BP in terms of gradient estimation. Additionally, the paper presents an activity-based variant (ANP) that estimates the gradient using differences between clean and noisy activations, thus bypassing the need for precise noise measurement. A key contribution is integrating a layer-wise input decorrelation mechanism, which mitigates the bias in NP updates and accelerates convergence.

Numerical experiments demonstrate that these modified NP algorithms, particularly when combined with input decorrelation, significantly enhance performance compared to standard NP and, in some cases, approach BP-level accuracy. The study also shows that these methods can be extended to noisy systems where the noise process is not directly observable, making them applicable to both neuromorphic computing and potential models of biological learning.

**Strengths:**

**Biological Motivation**: The exploration of node perturbation (NP) as an alternative to backpropagation is compelling due to its alignment with plausible biological mechanisms, negating the need for backward passes and allowing learning from reward signals. Previous work on NP suffers from poor performance and reliance on specific and accurate noise control. This work improved on previous studies by offering significant improvements that could potentially make NP a competitive framework.
- **Innovative Formulations**: The introduction of iterative node perturbation (INP) and activity-based node perturbation (ANP) adds theoretical depth, notably linking perturbation approaches with directional derivatives and improving the stability of NP in noisy environments. In particular, the authors show that the loss gradient can be computed without precise control over the noise process. This solution is elegant and informative.
- **Decorrelation Mechanism**: The incorporation of input decorrelation as an unsupervised learning mechanism demonstrates clear improvements in convergence speed, adding practical value to NP and its variants, while maintaining biological plausibility.

**Weaknesses:**

1.  **Scalability Limitations**: While the paper suggests that scaling to larger problems could be addressed through parallelization, this solution conflicts with the biological motivation emphasized throughout the text. The authors should reconcile this discrepancy by exploring biologically feasible alternatives or clarifying the practical biological implications.  Notably, this approach is clearly relevant for neuromorphic computing, particularly since the noise perturbations in this framework can be arbitrarily small.

2. **Gradient Approximation**: Theoretical analyses (e.g., Equation 4) focus on mean gradients. Still, the role of noise variance and the number of noisy samples in the stability and efficiency of gradient estimates are underexplored. Since the authors emphasize the framework's efficiency, claiming high performance can be achieved with a small number of noisy passes, the typical, rather than the mean loss gradient, should be analyzed.

3. **Decorrelation Analysis**: Adding an unsupervised learning rule for input decorrelation in each layer is an intriguing and potentially beneficial approach. However, the paper’s analysis of this aspect is insufficient, both numerically and theoretically. The improvement observed from decorrelating inputs, as demonstrated in Figure 3, is unsurprising. In a single-layer architecture, this step functions similarly to an additional linear transformation or data preprocessing step, which is expected to yield performance gains. This effect diminishes the novelty of the finding.
   Furthermore, while Figure 4 shows notable improvements in BP training accuracy with decorrelation, the minimal test accuracy gains suggest overfitting, indicating that the method primarily accelerates convergence without enhancing generalization. This point needs further exploration to determine the trade-offs between train and test performance. Moreover, Figure 4 highlights that input decorrelation does not enhance—and may even degrade—performance in convolutional networks. This discrepancy calls for a more thorough investigation into the conditions under which decorrelation aids or hinders performance. The authors should address these limitations and clarify whether decorrelation consistently benefits deeper and more complex architectures, or if its effectiveness is limited to simpler cases.

4. **Learning in the deep hidden layers**: Figure 2 indicates that the gradient alignment in the output weights closely matches that of BP, which may be due to the low-dimensional nature of the output space. This raises a concern that the observed performance, which falls short of BP’s, could be primarily driven by the readout weights, potentially bolstered by the unsupervised decorrelation step applied to the layer activities. This implies that the NP algorithms may contribute minimally to learning in the deeper layers.
   To address this issue, the authors should provide evidence that ANP/INP enhances learning throughout the network, rather than merely acting as a support for the final readout layer. Specifically, they should demonstrate that these algorithms outperform a simpler baseline approach involving unsupervised learning in the hidden layers followed by SGD or NP for training the output layer.
### Minor Comments
- **Appendix Clarifications**: The derivations in Appendix C do not add significant new insights beyond the main text and should be expanded to include formal proofs that strengthen the theoretical claims made in the main manuscript.
- **Clarification of Sample Averaging**: The use of a noise direction vector $v$ for each sample should be clearly articulated to explain how this averaging over noise directions ensures accurate gradient approximation.

**Questions:**

1. **Variance and Convergence of the Loss Gradient**: Could the authors provide an analysis of the variance in the estimated loss gradient as a function of the number of samples? Additionally, what is the typical convergence rate of the loss with an increasing number of samples? This data would offer valuable insights into the scalability and efficiency of the proposed methods.

2. **Impact of the Decorrelation Step**: Can the authors confirm that the observed performance improvements are not solely driven by the decorrelation rule? From Figure 4, it is unclear whether INP and ANP contribute significantly without the decorrelation step. What results would be obtained if only the decorrelation step was implemented, followed by NP applied solely to train the readouts?

While I have additional questions, addressing these principal concerns would be pivotal in reconsidering the rating of this work.

---

> ### Author Response · Authors · 2024-11-26
> **Response to reviewer's comments**
>
> We thank the reviewer for their careful and constructive feedback.
>
> __Regarding the scalability limitations__
> Though ANP might prove useful as an explanation for certain aspects of biological learning or as a template for learning in other noisy hardware, the experiment on Tiny ImageNet using multiple noise iterations was intended as a proof of concept, showing that, in principle, (D)INP converges to BP’s performance if given enough computation. Our claim is not that parallelization is plausible in biological networks.
>
> __Regarding gradient approximation__
> We have added an experiment to explore this topic. For NP, ANP and INP, we replaced Figure 2 with an experiment showing the alignment of the update with BP as a function of the number of noise samples. In Appendix G we also show these data as a function of the number of noisy forward passes.
>
> __Regarding decorrelation analysis__
> Though decorrelation can be seen as an extra preprocessing step in learning, note that the decorrelation is provided by a simple matrix multiplication, which is a linear operation. The decorrelated forward pass can therefore be expressed as y = Ax, where A is RW, with R the decorrelation matrix and W the forward weights. Therefore, the decorrelated matrix and forward matrices can be combined in inference and models therefore have the exact same parameterization as a regular forward model. The benefit of decorrelation does not come from more steps or parameters being added, but from a more efficient learning process from the decorrelated features space.
>
> BP’s modest test accuracy benefit from decorrelation in Figure 4 (top) might well be explained by BP already performing well without decorrelation, causing test accuracy to run into the limit of the network architecture. Note that Figure 4 (top) does show a large test accuracy benefit for the NP-style algorithms, suggesting that decorrelation does not simply lead to overfitting.
>
> __Regarding learning in deep layers__
> We would like to emphasize that learning in the 3 hidden layer networks was significantly better than learning in the single layer networks and performance for DINP was best in the 6 hidden layer network. Significant learning was also seen on the Tiny ImageNet task, using 5 hidden layers. These results make it implausible that only the output layer was learning significantly.
>
> __Regarding variance and convergence of the loss gradient__
> This issue is now explored more in the additional experiment, showing alignment with BP as a function of the number of noise samples in Figure 2.
>
> __Regarding impact of the decorrelation step__
> Though in the smallest networks performance of DNP, DANP, and DINP does not differ much, in the case using multiple noisy passes (Figure 5) and the Tiny ImageNet experiments (Figure 6) significant performance differences emerge. Moreover, these differences align with the theoretical properties of ANP and INP as well as with the alignment analysis in Figure 2, showing that ANP and INP provide benefit in addition to decorrelation.

---

> > ### Comment · Reviewer_smGL · 2024-11-29
> > **Response to the authors' rebuttal**
> >
> > 1. Biological plausibility: your answer is not satisfying, as it still means this method will not scale well in biological settings.
> > 2. Thank you for the additional analysis; it has improved my understanding. However, this does not exactly answer my question since performing many “epochs” is not the same as averaging over several noisy passes for each update. This is because the noise is within the nonlinearity.
> > 3. Decoration: I accept your arguments.
> > 4. Learning in the deep layer. I still think that the last layers do most of the learning. However, I agree that there is a noticeable difference in the shallower layers.
> >
> > I still think this work is interesting and above the threshold for acceptance. However, I am not yet convinced it is a strong accept.

---

### Official Review · Reviewer_nexo · 2024-11-04

**Soundness:** 2
**Presentation:** 3
**Contribution:** 2
**Rating:** 3
**Confidence:** 4

**Summary:**

In this manuscript, the authors propose an improved algorithm for node perturbation (NP) with two key modifications compared to the standard NP. First, the weight update is calculated using the total change in activity at each hidden node instead of the direct perturbation at that node. Secondly, the algorithm incorporates a decorrelation step at each layer to minimize noise correlations. The authors demonstrate numerically that decorrelation robustly enhances the performance of NP in deep neural networks trained on the CIFAR-10 dataset. They also show that using the total change in activity for updates outperforms the vanilla NP in convolutional neural networks.

**Strengths:**

The manuscript is clearly written and well-motivated. Moreover, the numerical experiments convincingly demonstrate that decorrelation improves the performance of NP.

**Weaknesses:**

Section 2.1.2, especially L127-L128, provides an impression that INP is required to make the update unbiased. However, the vanilla node perturbation is known to converge to the true backpropagation at the infinite sample limit if the noise added to each layer is independent of each other (Fiete et al., 2007; Hiratani et al., 2022).

On the contrary, the ANP update rule is biased against back-propagation. This can be demonstrated in a one-hidden layer non-linear network $y = W_2 f (W_1 x)$ with a loss function L(y).
Given perturbations $h v_1$, $h v_2$, at $h \to 0$ limit, the ANP update for the second layer is
$$\begin{eqnarray}
\Delta W_2
&=& \frac{1}{h} \langle (L(\tilde{y}) - L(y)) (v_2 + W_2 [f'(W_1 x) \odot v_1] ) f(W_1 x)^T \rangle
\nonumber \\\\
&=&   \langle (v_2 + W_2 [f' (W_1 x) \odot v_1] ) (v_2 + W_2 [f' (W_1 x) \odot v_1] )^T \rangle \frac{\partial L(y)}{\partial y}   f (W_1 x)^T
\nonumber \\\\
&=& (I + W_2 \text{diag} [ f'(W_1 x)^2 ] W_2^T) \frac{\partial L(y)}{\partial y}  f(W_1 x)^T
\end{eqnarray}$$
Because the true gradient is
$\frac{\partial L(y)}{\partial W_2} = \frac{\partial L(y)}{\partial y} f(W_1 x)^T$,
the ANP update rule above is biased against the true gradient, implying that the claims made in section 2.1 are inaccurate.

Given this bias, it is unclear why ANP achieves better alignment with backpropagation in Figure 2. Nevertheless, biased updates can sometimes facilitate faster learning, as discussed in Song, Millidge, et al., Nature Neuroscience, 2024. Therefore, the results shown in the bottom panel of Figure 4 are potentially interesting.

Regarding comparison with BP, empirical results are presented in a somewhat misleading way. A key limitation of NP is its need for a low learning rate, which means performance comparisons with BP should be conducted across a wide range of learning rates.

**Questions:**

Could you elaborate on the implementation details in convolutional networks, particularly regarding how weight sharing was handled?

---

> ### Author Response · Authors · 2024-11-26
> **Response to reviewer's comments**
>
> We thank the reviewer for their careful and constructive feedback.
>
> __Regarding bias in ANP and NP__
>
> Indeed Hiratani et. al. (2022) show NP to be unbiased, but only under certain assumptions which we find unrealistic, namely the assumption of a linear network, or infinitely small perturbations which would essentially linearize the system.
>
> Under more realistic assumptions of non-linearity and small but measurable perturbations, it can be shown that NP and ANP are both biased, but with ANP having a more local and interpretable source of bias. We demonstrate this based upon a Taylor Series expansion of a similar derivation as you have used and added an Appendix D to explain this. Ultimately, a Taylor expansion shows that in the case of a finite sized perturbation (which is always the case when implementing NP practically) updates under NP contain error terms related to downstream impacts of noise which ANP does not. Furthermore, in the case of decorrelated perturbation state differences, ANP is less biased.
>
> __Regarding the convolutional implementation__
>
> The convolutional implementation uses a built-in convolutional layer (e.g. from TensorFlow or PyTorch) and then adds noise to all of its outputs independently. To compute the weight update, image patches are extracted from the input using the same convolutional kernel dimensions as used in the layer object, after which the updates are first computed per patch and then averaged over patches.

---

> > ### Comment · Reviewer_nexo · 2024-11-27
> >
> > I appreciate the authors' efforts in revising the manuscript. However, the revised version still fails to address my primary concerns regarding its technical soundness.
> >
> > My main issue is that the proposed learning rule remains biased with respect to BP, even in the infinitesimal noise limit. This contrasts with the standard node perturbation (NP) rule, which is unbiased with respect to BP. While bias in the learning rule may not be inherently problematic, it is concerning that this significant limitation is not acknowledged in the manuscript. Instead, the authors claim their proposed rule is a "more principled approach to node perturbation" (Line 133) and has "a more solid theoretical foundation" (Line 67) without clear justification.
> >
> > Furthermore, the arguments presented in the newly added Appendix D are mathematically inaccurate. The authors assert that “there are terms involving the correlation between noise vector $v_2$ and higher order terms involving $v_1^⊤ v_1$ and the network activity Hessian. We cannot, in general, say anything about the unbiasedness of these terms and these are sources of unexplained error in existing work.” However, the term they reference, $\left\langle v_2 (v_1^T \nabla^2_{y_1} y_2 v_1)^T \right\rangle$, is zero if $v_1$ and $v_2$ are independently sampled zero-mean random variables. Therefore, it doesn’t induce bias.
> >
> > Their subsequent claim that “This can be brought to identity (or at least diagonal) by decorrelated activities and decorrelated activity differences,” is also inaccurate. As previously mentioned, assuming that perturbation vectors $v_1, v_2$ are white noise with amplitude $\sigma_v^2$, we have $\left \langle (\tilde{y}_2 - y_2) (\tilde{y}_2 - y_2)^T \right \rangle = \sigma_v^2 \left( I + W_2 \text{diag} [f' (W_1 x)^2] W_2^T \right) + \mathcal{O} (\sigma_v^4)$. Thus, $\left \langle (\tilde{y}_2 - y_2) (\tilde{y}_2 - y_2)^T \right \rangle$ is not diagonal in general.
> >
> > A secondary concern relates to the comparison with BP. As noted in my previous comments, a key limitation of NP is that it requires a small learning rate for convergence due to high variance (Werfel et al., NeurIPS 2003). Comparing NP with BP at a fixed small learning rate may give the misleading impression that the proposed methods (INP/ANP) are competitive with BP. In practice, BP is expected to achieve much faster convergence by using a larger learning rate. Therefore, the statement in the abstract claiming "large improvements in parameter convergence and much higher performance on the test data, approaching that of BP" is misleading.
> >
> > While the work offers some interesting contributions—such as increased biological plausibility by eliminating the need for neurons to track the source of noise and the idea of input decorrelation to improve NP—I believe the current manuscript lacks sufficient scientific rigor.

---

> > > ### Author Response · Authors · 2024-11-27
> > > **Response to reviewer's secondary comments**
> > >
> > > __Regarding bias in NP and BP__
> > > Thank you for your further comments and more detail on your perspective. In regards to our acknowledging of inherent bias in our ANP formulation relative to BP, we understand your concern. To mitigate this, we have modified our paper by adjusting the sentences which you indicated. Note that our claim of a ‘more principled approach’ on L133 and ‘solid foundation’ were both comments made in respect to (and in sections regarding) INP, which we resolutely stand by. INP does not require any consideration for the impact of propagated noise. However, we have modified our claims in the methods section on ANP to directly point out that:
> > > `However, relative to BP our derived ANP rule has a biased estimation in gradient measurement. Appendix D describes this bias and shows that, when considering finite noise injection, this bias is measurable in closed form and is interpretable.`
> > > and
> > >  ` … upon averaging across many samples can approximate the true gradient with a distinct bias based upon correlations in propagated noise.`
> > >
> > > Beyond this, our explicit naming of the Hessian term was perhaps unfair. We have updated the text to refer to `...beginning with the Hessian but also including third, fourth and higher derivative terms.` In general, one cannot say the induced bias by all of these terms is zero - especially given that even a simple non-linearity such as ReLU applied to random noise (with zero mean) will produce an output with non-zero mean.
> > >
> > > Furthermore, we acknowledge that $$\left\langle (\tilde{y}_2 - y_2)(\tilde{y}_2 - y_2)^T \right\rangle$$ is not diagonal in general. Our goal was to point out that its bias is well described, as opposed to an infinite series of Taylor expansions, and by inclusion of a decorrelation matrix (on top of a weight matrix W) one could try to optimize for this term to become diagonal. Text in the last sentence of  Appendix D has been updated to better reflect our intended message such that we now only claim `This is a much more interpretable bias term which could even be targeted directly to be made diagonal if a practitioner was interested in doing so.`.
> > >
> > > __Regarding learning rates for NP and BP__
> > > In this work, we actually optimized learning rates for all algorithms separately to ensure a fair comparison. For the single layer case, it just happened that the optimal learning rate was identical for BP and NP.

---

### Author Response · Authors · 2024-11-26
**Updates to paper based on reviewer's comments**

We again thank our reviewer's for their constructive feedback. Several changes have been made to the paper:

- The introduction now mentions some of the suggested literature regarding other bio plausible algorithms.
- Figure 2 has been replaced by a figure that measures the alignment of NP, ANP and INP updates w.r.t. BP as a function of the number of noise iterations, to give more insight into both the scaling properties of the algorithms and the degree to which theoretical properties of the algorithms play out in practice.
- An Appendix D has been added, providing an in-depth mathematical analysis showing ANP to be less biased than NP.
- An Appendix G has been added, showing the same data as Figure 2, but as a function of the number of noisy forward passes for each algorithm, essentially comparing their alignment per unit of computation.

---

### Meta-Review · Area_Chair_w5k3 · 2024-12-24

**Metareview:**

This submission addresses the important topic of exploring alternatives to backpropagation (BP) by investigating node perturbation (NP) frameworks with biologically plausible learning algorithms. While the paper has notable strengths, including a clear exposition of the methods and improvements to NP through iterative (INP) and activity-based (ANP) formulations, reviewers identified some weaknesses, see below.

**Additional Comments On Reviewer Discussion:**

Key weaknesses include unresolved concerns about the bias in the ANP update rule, which is claimed to provide a "principled approach" despite being theoretically less aligned with BP than standard NP under certain conditions. Furthermore, the empirical results, while promising, lack robust validation against alternative learning rates and stronger baselines, raising questions about the generalizability of the findings. The claims around biological plausibility remain speculative without concrete connections to biological mechanisms.

---

### Decision · Program_Chairs · 2025-01-22

Reject